# Neural correlates of error-monitoring and mindset: Back to the drawing board?

**Tieme W. P. Janssen** [1] *, **Smiddy Nieuwenhuis**[1], **Jamie Hoefakker** [1], **Patricia D. Dreier Gligoor**[1], **Milene Bonte**[2], **Nienke van Atteveldt**[1]

**1** Department of Clinical, Neuro- & Developmental Psychology, Faculty of Behavioural and Movement sciences, Vrije Universiteit Amsterdam, Amsterdam, The Netherlands, **2** Department of Cognitive Neuroscience, Faculty of Psychology and Neuroscience, Maastricht University, Maastricht, The Netherlands

* twp.janssen@vu.nl

## Abstract

The different ways students deal with mistakes is an integral part of mindset theory. While previous error-monitoring studies found supporting neural evidence for mindset-related differences, they may have been confounded by overlapping stimulus processing. We therefore investigated the relationship between mindset and event-related potentials (ERPs) of error-monitoring (response-locked Ne, Pe), with and without overlap correction. In addition, besides behavioral measures of remedial action after errors (post-error slowing and accuracy), we investigated their neural correlates (stimulus-locked N2). Results indicated comparable Ne, but larger Pe amplitudes in fixed-minded students; however, after overlap correction, the Pe results were rendered non-significant. A likely explanation for this overlap was a near-significant effect of mindset on the preceding stimulus P3. Finally, although N2 was larger for trials following errors, mindset was unrelated. The current study shows that the relationship between error-monitoring and mindset is more complex and should be reconsidered. Future studies are advised to explore stimulus processing as well, and if needed, to correct for stimulus overlap. In addition, contextual influences on and individual variation in error-monitoring need more scrutiny, which may contribute to refining mindset theory.

**Data Availability Statement:** Data are archived on DataverseNL: https://doi.org/10.34894/IRCBS8.

**Funding:** NvA received an European Research Council Starting Grant (grant number: 716736).

## Introduction

Making mistakes is inherent to learning, and one could say, essential to learning. Students, however, may hold different views on what these mistakes mean to them. Some will interpret mistakes in a constructive manner, signaling the need for more learning and effort (mastery-oriented), while others tend to interpret mistakes as personal failure and avoid them if possible (helpless-oriented). Past research shows that these two kinds of students may have different ability beliefs [1], the former believing their abilities are malleable with effort (growth mindset), while the latter believing their abilities are fixed entities (fixed mindset). So far, only a few neurocognitive studies have investigated the neural mechanism underlying mindset [2], typically by investigating error and feedback processing. These studies yielded some common

URL: https://erc.europa.eu/funding/starting-grants.
The funders had no role in study design, data
collection and analysis, decision to publish, or
preparation of the manuscript.

**Competing interests:** The authors have declared
that no competing interests exist.

findings, but also inconsistencies, open questions and methodological limitations. Neuroscience research has the potential to refine theoretical models of non-cognitive skills, such as mindset, and may contribute to developing well-informed interventions [3]. This is especially important considering recent discussions about the foundations of mindset theory [4] and the relevance of mindset interventions [5, 6].

With the event-related potential (ERP) technique two components have been identified to map error-monitoring, the error negativity (Ne) and the error positivity (Pe). Ne is a response-locked ERP that reaches a negative fronto-central maximum around 50-100ms after the initiation of an erroneous response [7], and is presumably generated in the anterior cingulate cortex [ACC; 8]. This is followed by a positive centro-parietal deflection, Pe, approximately 200-500ms after the error. Where Ne is thought to be a more automatic process, Pe reflects conscious or emotional evaluation of the error [9]. Unlike the Ne, the neural generators of the Pe are less well known, and are presumably more distributed [10], including the ACC [11], posterior cingulate cortex [12, 13], and the parietal and insular cortices [14, 15]. Post-error slowing (PES) is a commonly used behavioral metric for error-monitoring, which describes the remedial action of people slowing down in the subsequent trial after committing an error [16]. Although this should increase post-error accuracy (PEA), because of a speed-accuracy tradeoff, this is not consistently found [17]. Ne and Pe during error trials have both been related to PES during the following trial [10].

Relatively few studies have focused on the neural mechanisms that underly PES itself. Interestingly, Chang et al. [18] found the enhancement of N2 amplitude to be a neural correlate of PES. N2 may reflect participants' strategic adjustment following an error, involving frontal cognitive control mechanisms. This would be in line with two fMRI studies demonstrating increased activation of the ventrolateral prefrontal cortex (VLPFC) during PES [19, 20]. The VLPFC has been implicated in mediating attention and action reversal [18], which corresponds to what happens when a prepotent response is reversed and reaction times increase after an error is made.

Currently, four ERP studies have investigated mindset, of which three focused on error-monitoring (Ne, Pe) and one on feedback processing [21–24]. Error and feedback processing can be seen as two variants of feedback processing, one concerned with processing of internal and the other with external feedback [25]. While mindset was not related to Ne, all three error-monitoring studies found a relation between mindset and Pe, the more conscious component of error monitoring. The direction of the association, however, seems to depend on whether mindset was measured as *trait* in children [21] and adults [22]—with larger Pe in growth-minded participants—or *induced* with a brief manipulation in adults—with larger Pe in the fixed-mindset group [23]. In the first two studies [21, 22], PEA but not PES was higher in students with a growth mindset. In adults, Pe mediated the relation between mindset and PEA. These combined findings suggest that when measured as a trait, growth-minded individuals have enhanced error awareness, which may in turn facilitate remedial action after the errors.

This conclusion is corroborated by a resting-state fMRI study [3], which demonstrated growth mindset to be related to increased connectivity between brain regions that support regulation strategies and error monitoring, comprising the dorsal striatum, dorsal ACC and left dorso-lateral prefrontal cortex (DLPFC). Another fMRI study relevant to mindset, showed mastery-oriented participants to have relatively increased DLPFC activity during negative feedback compared to performance-approach-oriented participants [26].

More difficult to reconcile are the opposite mindset induction findings, with larger Pe amplitudes in the fixed mindset condition, irrespective of accuracy [23]. In contrast, for the growth mindset condition, larger P3 amplitudes were found for stimuli *preceding* the erroneous responses. These findings are in line with the distinction between performance goals

(focused on outcome; Pe) that are more common in fixed-minded students, and learning goals (focused on the stimulus; P3) that are more common in growth-minded students. Moreover, in another study [24], adults endorsing a fixed mindset and performance goals demonstrated larger P3 amplitudes for negative performance feedback after errors. Taken together, these results suggest that fixed-minded students interpret performance feedback, either internal or external, to be more salient or even threatening.

A limitation of error-monitoring studies is the potential overlap of stimulus- and response-related processing, due to the adjacency between stimulus and response, which may confound the response-locked error-monitoring ERPs. In fact, Pe may even be a delayed P3 related to stimulus processing, rather than error-monitoring [27]. Especially problematic is that the stimulus is usually a NoGo or Stop stimulus in inhibition paradigms, inducing similar frontal cognitive control processes as error-monitoring (e.g. anterior cingulate). This is an important limitation to address, as it can fundamentally change the interpretation of neurocognitive mechanisms related to mindset. Moreover, as discussed, Schroder et al. [23] found mindset to affect stimulus P3 preceding errors, which may have confounded the Pe results. In the other two studies [21, 22], stimulus processing preceding errors was not explored and therefore potential confounding effects on the error-monitoring ERPs are unknown.

In the current study with young adult students, we investigated the relationship between mindset and event-related potentials (ERPs) of error-monitoring (response-locked Ne, Pe). In addition, besides behavioral measures of remedial action after errors (PES and PEA), we investigated their neural correlates (stimulus-locked N2). N2 is presumed to reflect cognitive control mechanisms, involving action reversal. Note that the neural correlates during remedial action are still unexplored in relation to mindset. We hypothesized larger Pe and N2 amplitudes and higher PEA in growth-minded students, based on the most similar studies [21, 22]. Importantly, we repeated the error-monitoring analysis, while correcting for *preceding* stimulus overlap, to test the hypothesis that Pe results are driven by differences in stimulus processing. The results of the corrected error-monitoring analysis were deemed more reliable and are therefore discussed as final results.

## Materials and methods

### Participants

In total, 116 psychology or pedagogy bachelor students participated in this study. The first 20 were excluded due to an unintended programming setting in the stop-signal task; stop stimuli never followed each other directly. This was problematic, because participants could become aware of this and take a more risky style of responding directly after each stop trial (knowing that no inhibition is needed). Indeed we found evidence for this in a previous publication [28]. Another three had missing EEG data, two received incorrect instructions and two had already started their masters. Of the final sample (N = 89), thirty-three were native-Dutch-speaking, while 56 were exchange students receiving English instructions. Participants were compensated with either course credits or 20€. We planned a minimum of 65 participants, based on a power-analysis in G\*power with an expected medium effect size (f2 = .20), alpha (0.05) and power (0.80).

### Procedure

The study was conducted according to the Declaration of Helsinki, and was approved by the local ethics committee of the Faculty of Behavioural and Movement Sciences at the Vrije Universiteit (Amsterdam). All students signed informed-consent before participating in this study. The EEG study was preceded, at least one week, by an online questionnaire. The EEG

study started with resting state recordings, followed by an arithmetic task (15 minutes) and the stop-signal task (SST; 25 minutes). The current article focuses only on the SST.

### Behavioral assessment

Mindset was measured using the revised self-theory scale designed by De Castella & Byrne [29]. This questionnaire consists of eight items. Each item was scored on a Likert scale from 1 (strongly disagree) to 6 (strongly agree). The scores on the four fixed mindset items were reversed and added to the scores of the four growth mindset items to create one mindset scale. Higher sum scores reflected greater growth mindset endorsement. Internal consistency was high ($\alpha$ = .93).

### Stimuli and task

The SST involved two types of trials: go trials (G) and stop trials (S). G trials contained left or right pointing airplanes (go stimuli), requiring either a left or right button response. Each trial started with a white fixation cross, centered on a black background for 500ms, followed by a go stimulus for 1250ms and a black screen for 650ms. Inter-trial-intervals varied randomly between 0, 50, 100, 150 and 200ms. In a randomly selected 25% of the trials, go stimuli were followed by a visual stop stimulus superimposed on the go stimulus, requiring the participants to withhold their response (S trials). The delay between the go and stop stimulus (stop stimulus delay, SSD) varied trial-by-trial using a tracking algorithm which increased or decreased the delay with 50ms, depending on whether or not the previous stop trial resulted in stop success (SS), yielding approximately 50% stop errors (SE; unable to withhold response after stop stimulus). Because the error rate can be controlled, resulting in sufficient and comparable error rates among participants, the SST is a frequently used task to investigate error-processing.

Two practice blocks and 6 experimental blocks of 100 trials were administered in 25 minutes with the trials presented in a fixed pseudorandomized order. Participants were instructed to respond both quickly and accurately to the go stimuli and withhold their response when a stop stimulus was presented. Analysis of task performance was based on Chang et al. [18]. G trials were further divided based on the prior trial. For example, pSE (post stop error) trials were correct go trials that were preceded by a SE trial, while pG (post go) trials were correct go trials that were preceded by another correct go trial. Dependent variables were: reaction time (ms) for the different trial types, post-error slowing (PES; RT pSE-pG), accuracy (%) for pG and pSE, and post-error accuracy (PEA; accuracy pSE-pG). See Fig 1A for an overview of trial types.

### Electrophysiological recordings

Continuous EEG was recorded at 512Hz using the ActiveTwo Biosemi system (Biosemi, Amsterdam, The Netherlands) from 128 electrodes and an Electro-oculogram (EOG) was obtained. Off-line analyses were performed with Brain Vision Analyzer 2 (Brain Products, Gilching, Germany, Version 2.1). A band-pass filter of 0.1–30 Hz (24 dB/oct) was applied and scalp electrodes were re-referenced to the average of the mastoids. Broken electrodes were interpolated with the spherical splines method [30]. Ocular artifacts were estimated and corrected with semi-automatic independent component analysis [31], and automatic artifact rejection was applied to segments based on the following criteria: maximum allowed voltage step of 50μV/ms, maximal peak-to-peak amplitude difference of ±150μV, and minimal low activity of 0.50μV for 100ms intervals.

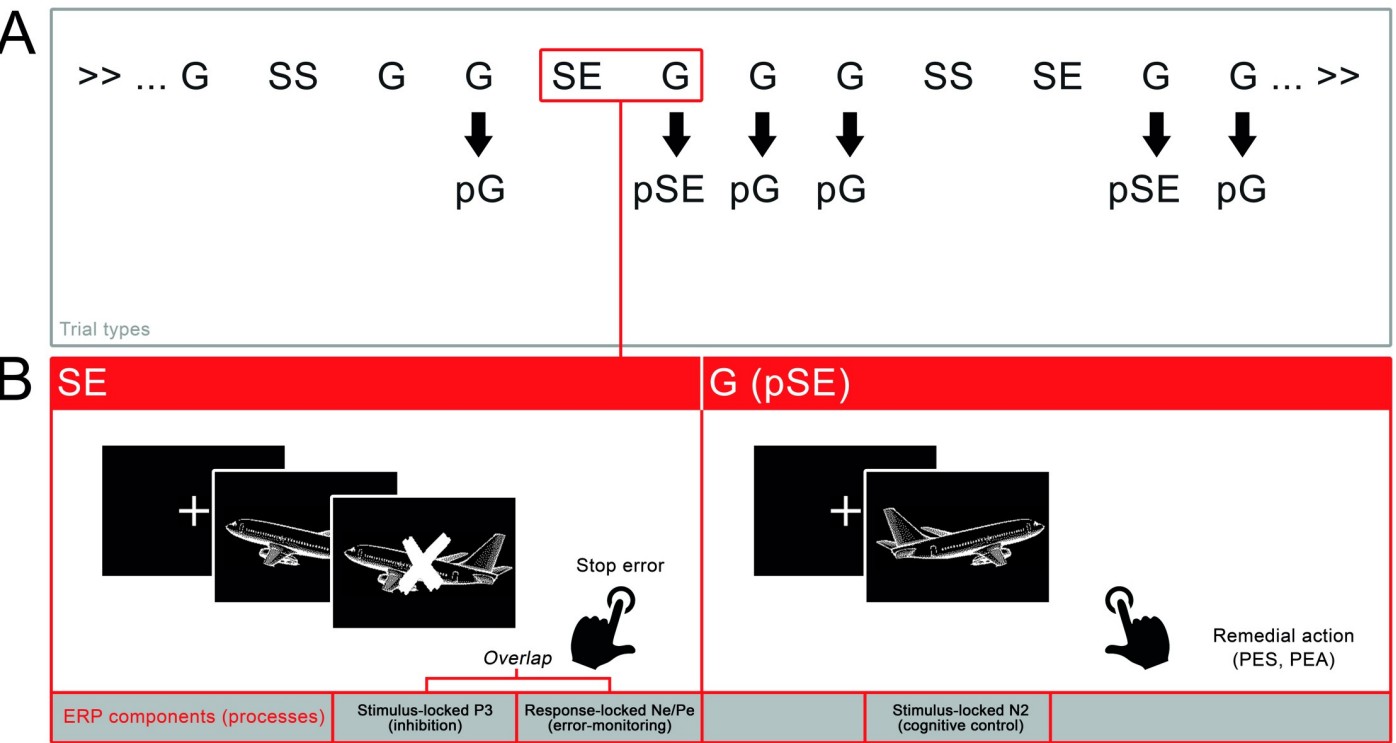

**Fig 1. Overview of trial and ERP types and the three related consecutive processes (1) inhibition, (2) error-monitoring and (3) cognitive control.** *Note*. Trial types: G = go, SS = stop success, SE = stop error, trial types were subsequently divided based on what kind of trial preceded them, indicated with 'p' for post. ERP analyses focused on SE and G (pSE) trial types.

### Segmentation

The distinction of the various trial types allowed us to analyze three different consecutive processes and their ERP correlates: (1) inhibition, (2) error-monitoring and (3) cognitive control, see Fig 1B. The first analysis focused on the stop-*stimulus-locked* P3 component, the second analysis focused on the *response-locked* Ne and Pe components, while the third analysis focused on the go *stimulus-locked* N2 component.

**Inhibition.** Trials were segmented from -100ms to 800ms relative to the stop stimulus. Subsequently, a 100ms pre-stimulus baseline was applied and averages were obtained for SS. In the current study, we focused on the potential confounding overlap of the first (inhibition) on the second process (error-monitoring). The ERP induced by the stop stimulus (stop ERP) was therefore used for overlap correction, as described in the next paragraphs. To get a further indication of how problematic the overlap is for the subject of this study, we related mindset to the stop P3 component.

**Error-monitoring.** First, both types of response trials, SE (stop error, incorrect response) and G (go, correct response), were segmented from -200 to 600 ms relative to the button press (response-locked), baseline-corrected for the interval -100 to 0 ms and artefact-free segments were averaged to obtain ERPs (Ne and Pe).

**Error-monitoring (corrected for overlap).** Due to the adjacency between the stop stimulus and response during SE trials, inhibition-related processing induced by the stop stimulus may overlap with error-related processing induced by the response. Therefore, a validation analysis was performed by removing the confounding overlap with ADJAR [32].

With ADJAR, for each participant a previous event distribution (stop-stimulus) is determined relative to the timing of the current event (response error). In other words, the current

event is fixed in time, while the previous event varies in timing. Subsequently, a convoluted waveform is calculated by averaging the participants' successful stop ERP over the range of the event distribution. This convoluted waveforms is then extracted from the current event waveform (response-locked ERP), removing the overlap of the previous event. The result is mostly a low-pass filtering, as high-frequency components cancel each other out. For a theoretical discussion about convolution, see Luck [33].

**Cognitive control.** First, pG and pSE G-trials were segmented from -200 to 800 ms relative to the onset of the Go stimulus (stimulus-locked), baseline-corrected for the interval -100 to 0 ms, and artefact-free segments were averaged to obtain ERPs (N2).

## ERP components

Based on all 89 participants, irrespective of their reported mindset, grand average ERPs, scalp topographies and difference waves for each trail type were inspected to define analysis windows (see Figs 2, 3 and 6). In addition, characteristics of the ERP waveforms were compared to the literature [18, 34–36], to ascertain the correct identification of ERP components. This resulted in the following time windows: inhibition P3 (336-386ms), error monitoring Ne (77-127ms) and Pe (203-403ms), cognitive control N2 (217-276ms) and, unplanned, P3 (310-410ms). The median split in Figs 2 and 3, with a growth and fixed mindset group, is only for visualization purposes and did not inform the selected time windows.

More specifically, all windows were symmetrically centered based on the latency of the electrode with the largest amplitude. Ne and Pe windows were based on difference waves (incorrect-correct response). Mean voltage amplitudes were extracted and used for statistical analyses. To validate the condition effects for error-monitoring, besides exporting Ne and Pe difference waves, also the Nc/Pc [correct] and Ne/Pe [incorrect] were exported separately. A minimum of 20 artefact-free segments were required for each condition to be included. Due to the task design, there were more correct than incorrect responses (SE) contributing to their respective ERPs, resulting in different signal-to-noise ratio's. However, when using mean rather than peak amplitudes, it will not bias the results and it is recommended to retain all trials to decrease Type II error rate [37].

## Statistical analysis

Statistical analyses were performed with SPSS 25 (IMB, Armonk, NY). Significance was assumed if $p < .05$ (two-tailed). Effect sizes are reported as partial eta-squared ($\eta_p^2$), with effects interpreted as small (.01), medium (.06) or large (.14). All analyses were first performed without mindset as covariate to validate the expected main condition effects. Subsequently, mindset was inserted as covariate. The mindset scale was log-transformed to obtain a normal distribution. Only for displaying purposes, mindset was dichotomised into a fixed and growth mindset group (median split).

**Behavioural.** Post-error reaction time slowing (PES) and accuracy (PEA) were tested with repeated measures ANOVAs, using one within-subject factor Condition (pG, pSE). All performance variables were log-transformed to normalize the data.

For all ERP analyses, multivariate test statistics are reported, a method known to be robust against violations of sphericity [38]. For purposes of data reduction, only midline electrodes Fz, Cz and Pz were included as within-subject Location, which captured all ERP topographies adequately (see Figs 2, 3 and 6). Another within-subject factor Condition was included for the inhibition analysis (stop success versus stop error; SS and SE), error-monitoring analysis (correct and incorrect response) and cognitive control analysis (post go versus post stop error; pG,

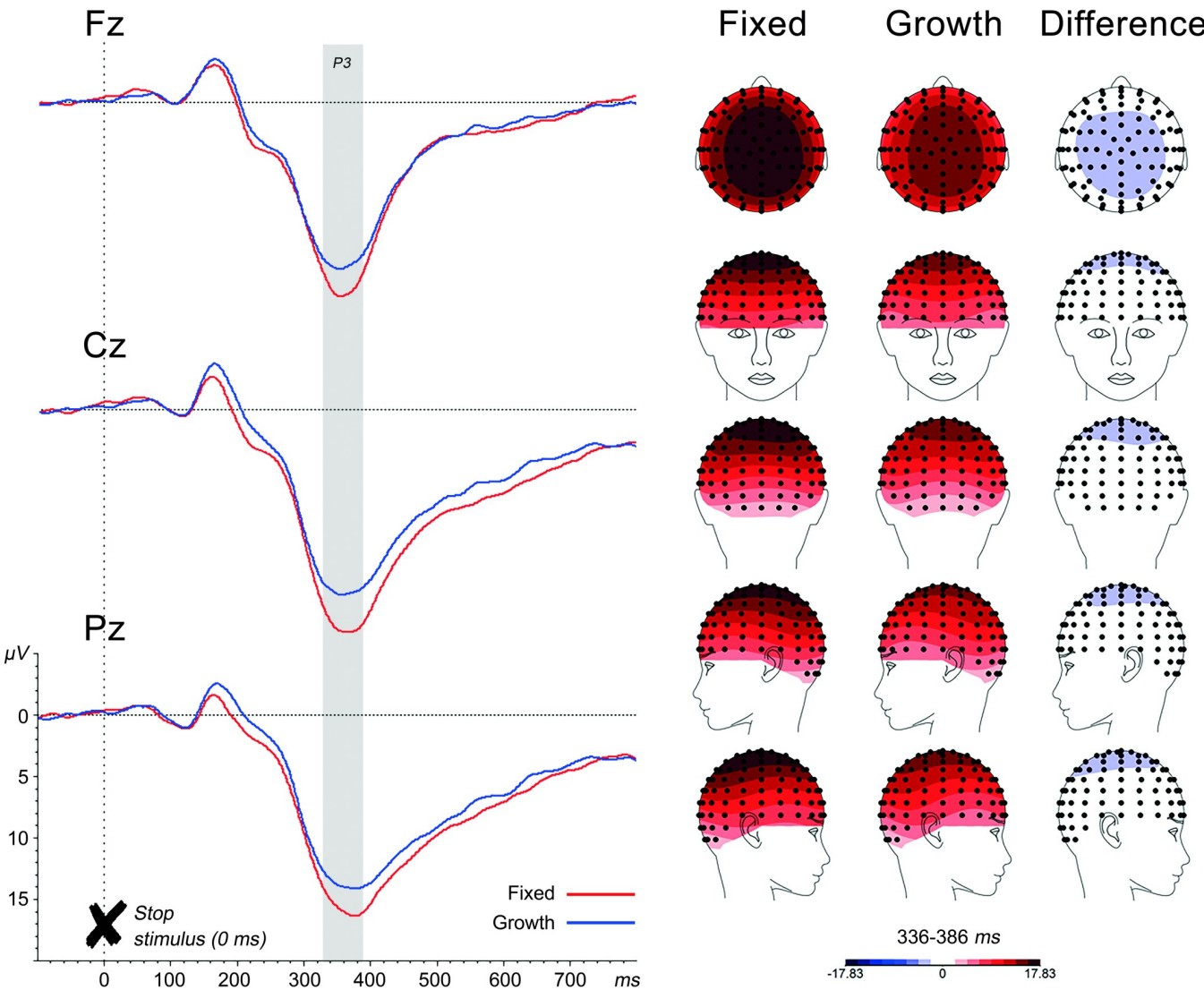

**Fig 2. Stimulus-locked ERPs for stop stimuli resulting in successful stop (SS) at midline electrodes with topographic maps of the P3 component.** *Note.* For display purposes, a median split was used to show fixed and mindset ERPs and P3 topographic maps. Gray area is the P3 analysis window.

pSE). Pearson correlations were performed between mindset and ERPs for each separate condition in the case of significant interactions.

## Results

### Performance and questionnaire outcomes

See Table 1 for means, SD and range. In the post-error slowing (PES) analysis, there was a medium/large effect of Condition (pG, pSE) on RT, $F(1,88) = 8.84$, $p = .004$, $\eta p^2 = .09$. PES was on average 10.7ms. Introducing mindset as covariate did not result in a significant interaction with Condition, $F(1,87) = 0.39$, $p = .535$, or a main effect of mindset, $F(1,87) = 0.58$, $p = .449$. Accuracy data (PEA) were highly negatively skewed, with a clear ceiling effect around 100% accuracy. Therefore, a non-parametric Friedman test was performed, which did not demonstrate significant Condition (pG, pSE) effects, $\chi^2(1) = 3.20$, $p = 0.074$. Mindset and PEA were not significantly correlated, $r(87) = .04$, $p = .697$.

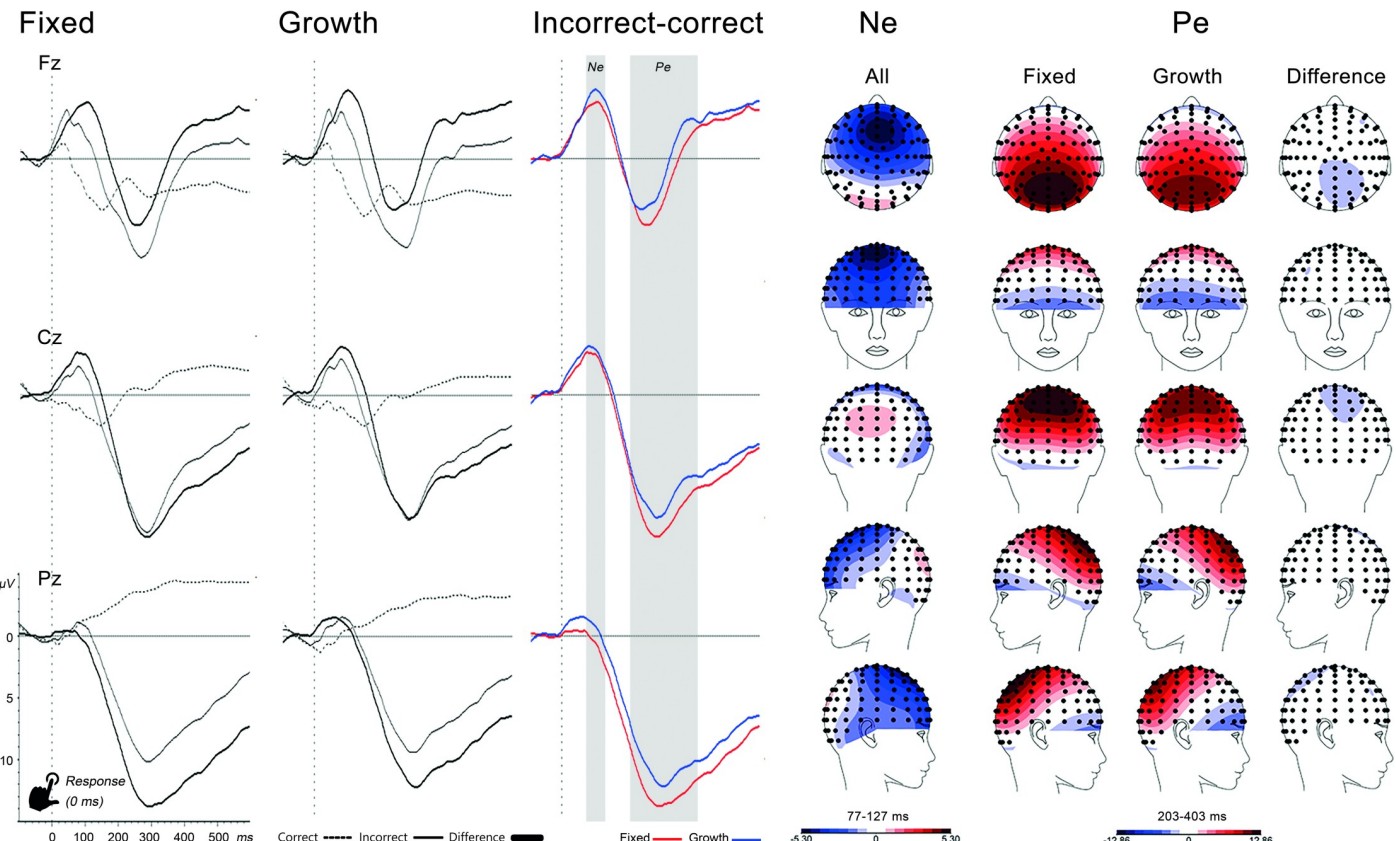

**Fig 3. Response-locked ERPs for correct, incorrect (stop errors) and difference ERPs at midline electrodes with topographic of the difference wave Ne and Pe components.** *Note*. For display purposes, a median split was used to show fixed and mindset difference-ERPs and Pe topographic maps. Gray areas are the Ne and Pe analysis windows.

### ERP components

**Data quality.** Average number of included EEG segments for ERP averaging were: SS (70), SE (63), correct response (426), incorrect response (64), pG (296), pSE (58). Minimal number of segments was 34. Average number of interpolated electrodes was 2.6. None of the quality indices were related to mindset (all $p > .338$).

**Inhibition.** See Fig 2 for stop-stimulus-locked ERPs at midline electrodes and topographies for the SS condition, separately for the fixed and growth mindset groups, based on a median split for display purposes. Statistical analyses are based on mindset as (continuous) covariate.

*P3 (336–386 ms)*. Both timing and polarity are in line with the literature [34]. There was a main effect of Location, $F(2,87) = 24.37$, $p < .001$, $\eta p^2 = .36$. Post-hoc tests showed maximal amplitude at Cz compared to Fz, $F(1,88) = 24.06$, $p < .001$, and Pz, $F(1,88) = 12.26$, $p = .001$. There was no effect of Condition (SE, SS), $F(1,88) = 0.10$, $p = .749$. Introducing mindset as covariate resulted in a near-significant main effect of Mindset, $F(1,87) = 3.83$, $p = .054$, $\eta p^2 = .04$, with larger P3 amplitudes in participants endorsing more of a fixed than a growth mindset, irrespective of condition (SE or SS).

**Error-monitoring.** See Fig 3 for response-locked ERPs at midline electrodes for correct and incorrect responses, separately for the fixed and growth mindset groups, based on a median split for display purposes. Topographic maps are based on difference waves.

**Table 1. Questionnaire and performance data.**

| | N = 89 | | |
| --- | --- | --- | --- |
| | *M* | *SD* | **Range** |
| Demographics | | | |
| Age | 20.9 | 1.9 | |
| Male/female | *21/68 female* | | |
| Questionnaire | | | |
| Mindset | 35.6 | 6.9 | 13–48 |
| Stop-Signal Task | | | |
| SE (RT) | 572.9 | 107.3 | 394–1052 |
| pG (RT) | 608.8 | 119.8 | 418–1095 |
| pSE (RT) | 618.8 | 124.9 | 419–1140 |
| PES (pSE-pG) | 10.7 | 28.1 | -47-93 |
| pG accuracy (%) | 98.7 | 1.5 | 89–100 |
| pSE accuracy (%) | 98.4 | 3.2 | 76–100 |
| PEA (pSE-pG) | -0.3 | 2.2 | -13-3 |

*Note*. Higher scores on Mindset indicate higher growth mindset; SE = stop error, pG = correct go trials that were preceded by another correct go trial; pSE = were correct go trials that were preceded by a SE trial; PES = post-error slowing; PEA = post-error accuracy, RT = reaction time

*Ne (77–127 ms)*. Timing, polarity, topography and condition effects were in line with the literature [35]. There was a Condition*Location effect, $F(2,87) = 60.60$, $p < .001$, $\eta p^2 = .58$. Posthoc tests showed higher Ne amplitudes during incorrect responses at Fz, $F(1,88) = 105.57$, $p < .001$, and Cz, $F(1,88) = 80.48$, $p < .001$, but not at Pz, $F(1,88) = .00$, $p = .976$. Introducing mindset as covariate did not result in any significant interaction or main effect (all $p > .15$).

*Pe (203–403 ms)*. Timing, polarity, topography and condition effects were in line with the literature [36]. Condition effects were dependent on scalp location as indicated by a Condition*Location interaction, $F(2,87) = 105.10$, $p < .001$, $\eta p^2 = .71$. Posthoc tests showed higher Pe amplitude during incorrect than correct responses at all three electrodes, with the largest effect sizes at Pz, $F(1,88) = 370.83$, $p < .001$, $\eta p^2 = .81$, and Cz, $F(1,88) = 208.70$, $p < .001$, $\eta p^2 = .70$, but a smaller effect size at Fz, $F(1,88) = 7.61$, $p = .007$, $\eta p^2 = .08$. With mindset as covariate, there was a Mindset*Condition interaction, $F(1,87) = 4.19$, $p = .044$, $\eta p^2 = .05$. Based on the topographic maps, which showed maximum amplitude differences at Pz, we conducted three separate correlations between mindset and each condition at Pz (Pc[correct], Pe[incorrect] and difference wave) to further explore the interaction. Mindset was positively related to Pc amplitude, $r(87) = .260$, $p = .014$, not significantly related to Pe, $r(87) = -.189$, $p = .076$, and negatively related to the difference wave, $r(87) = -.262$, $p = .013$. See Fig 4 for scatterplots and a median split display of the effects. In contrast to our hypothesis, Pe difference amplitudes were larger in participants who endorsed more of a fixed mindset, although the correlation with Pe (without subtracting Pc) did not reach significance. Another unexpected finding was higher Pc amplitudes in participants endorsing more of a growth mindset.

**Error-monitoring (corrected for stimulus overlap).** See Fig 5 for the ADJAR-corrected ERP waveforms. Due to space limitations, only effects of mindset on the corrected ERP's will be reported.

*Ne (100–150 ms)*. Compared to the uncorrected waveforms, the Ne peak amplitude was 33ms later and larger, while the topography was similar. The Mindset*Condition interaction was not significant, $F(1,87) = 2.39$, $p = .126$, $\eta p^2 = .03$.

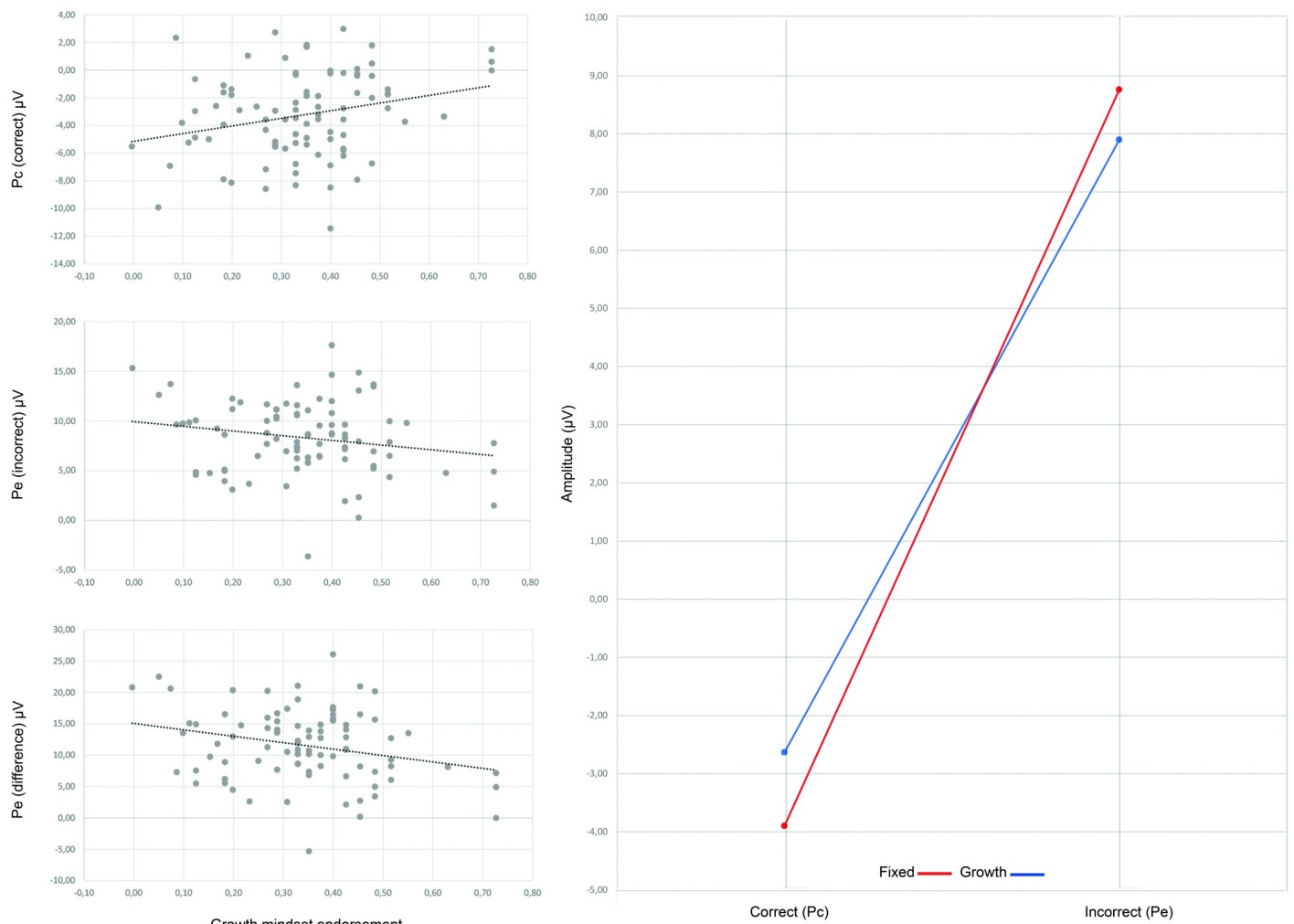

**Fig 4. Scatter plots of Pc/Pe/difference amplitude at Pz against growth mindset endorsement.** *Note*. The x-axis shows the log-transformed mindset scale, with more positive numbers reflecting stronger growth mindset, while lower numbers reflecting stronger fixed mindset endorsement. Dotted line is the regression line. Correct = correct response, Incorrect = incorrect response (stop error). Pc = positive correct ERP, Pe = error positivity ERP, Pe difference = Pe–Pc.

*Pe (347–547 ms)*. Compared to the uncorrected waveform, the Pe difference wave peak was less pronounced, later and smaller, while the topography was similar. In contrast to the uncorrected analysis, the Mindset*Condition interaction was no longer significant, $F(1,87) = 2.51$, $p = .117$, $\eta p^2 = .03$.

**Cognitive control.** See Fig 6 for ERPs at midline electrodes for go-trials in pG and pSE conditions and topographic maps for the N2 component.

*N2 (217–276 ms)*. Timing, polarity, topography and condition effects were in line with the literature [18]. There was a Condition*Location interaction, $F(2,87) = 5.29$, $p < .001$, $\eta p^2 = .13$. Posthoc tests demonstrated higher N2 amplitudes for pSE compared to pG at Cz, $F(1, 88) = 14.33$, $p < .001$, and Pz, $F(1,88) = 18.70$, $p < .001$, but not at Fz, $F(1,88) = 3.14$, $p = .080$. There were no significant interactions or main effects of mindset when included as covariate (all $p > .09$).

*P3 (310–410 ms)*. This was an unplanned analysis; however, the ERP waveforms hinted at possible Condition effects for this component. There was a Condition*Location effect,

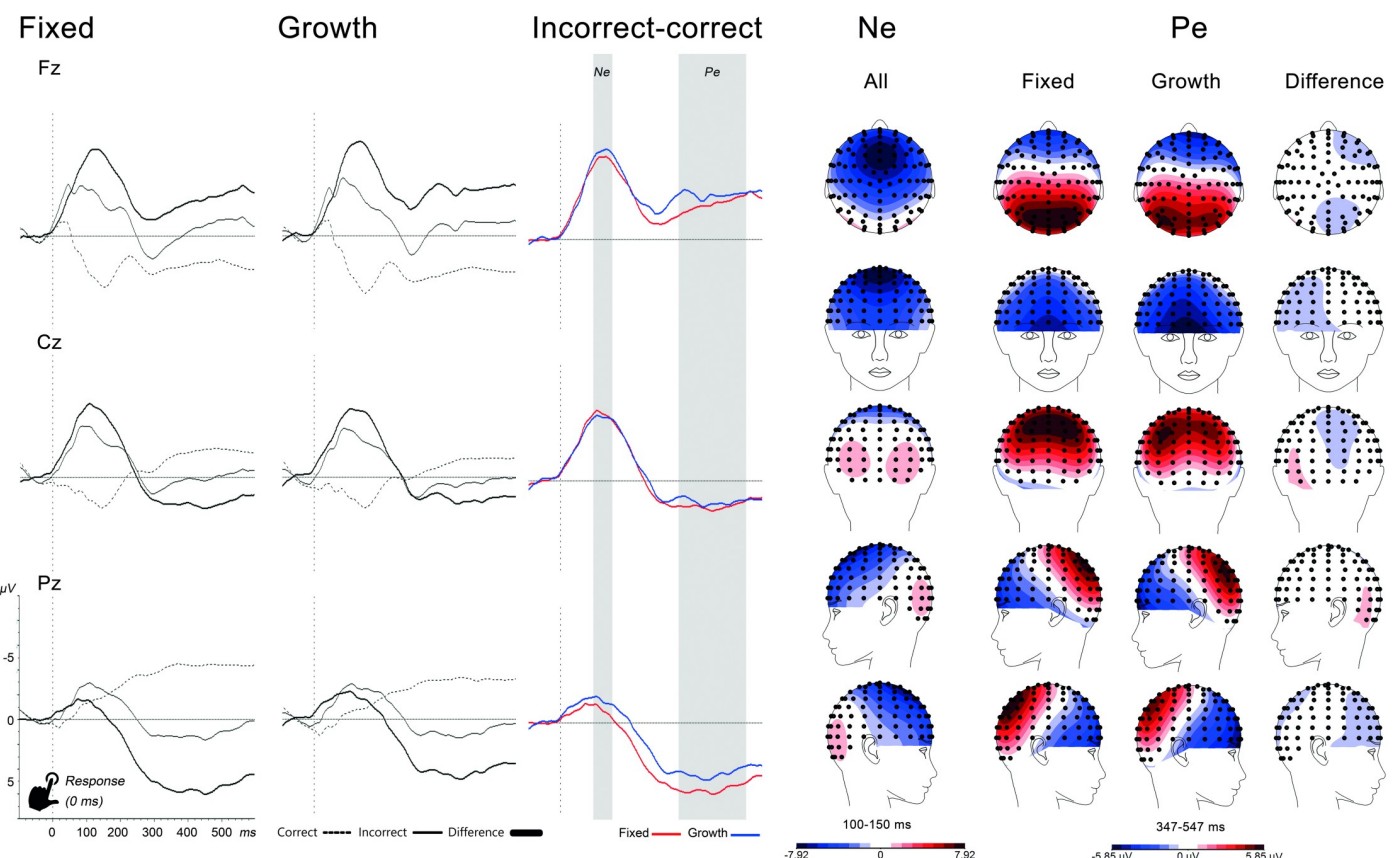

**Fig 5. Response-locked ERPs for correct, incorrect and difference ERPs corrected for stimulus overlap with ADJAR.** *Note*. For display purposes, a median split was used to show fixed and mindset ERPs and Pe topographic maps. Gray areas are the Ne and Pe analysis windows.

$F(2,87) = 4.15$, $p = .019$, $\eta p^2 = .09$. Posthoc tests showed condition effects for Cz and Pz ($p <$ .001), but not Fz ($p = .101$), with smaller P3 amplitude for pSE compared to pG. There were no significant interactions or main effects of mindset when included as covariate (all $p > .14$).

## Discussion

We aimed to better isolate event-related potentials (ERPs) of error-monitoring and their relationship with mindset, and in addition, to investigate the neural mechanisms of cognitive control in trials following errors. Therefore, three consecutive processes were investigated: (1) inhibition, related to the stop-stimulus before the error [P3], (2) error-monitoring [Ne, Pe] and (3) cognitive control [N2 and behavioral measures PES/PEA]. For the first process, we found a near-significant relation between mindset and P3 ($p = .054$), with larger P3 for those endorsing more of a fixed mindset. Considering the close proximity between stop-stimulus- (P3) and response- (Pe) locked ERPs and resulting overlap, the latter may be confounded. In other words, the Pe may be a delayed P3 related to stimulus processing. We found evidence in support of this hypothesis, as the uncorrected Pe analysis demonstrated a significant effect of mindset, which was rendered non-significant after correcting for overlap of the stop-stimulus ERP. Finally, mindset was not related to the third process, cognitive control as indexed by neural (N2) and behavioral measures (PES/PEA), in trials following errors.

First, we will discuss the findings pertaining to the first process, response inhibition, which involved stimulus processing preceding errors. Only one other study [23] investigated the

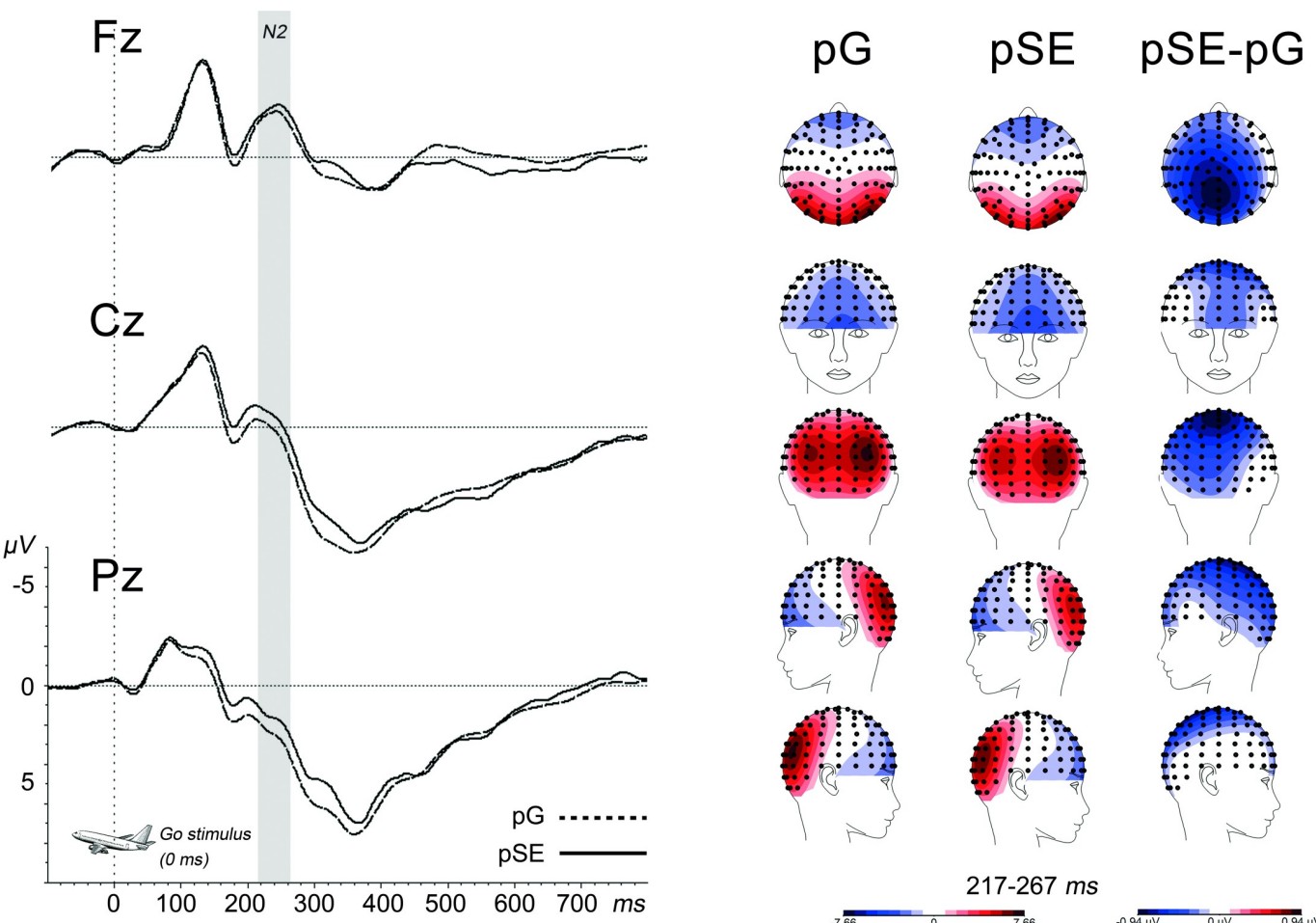

**Fig 6. Stimulus-locked ERPs for go stimuli following go (pG) or stop error (pSE) trials at midline electrodes with topographic maps of the N2 component.**
*Note.* Gray area is the N2 analysis window.

relation between mindset and stimulus-related processing, in addition to error-related processing. In this study either a growth or fixed mindset was induced using a brief manipulation involving the reading of a scientific article, with opposite results to the current study: they found larger P3 amplitudes in the growth mindset group. The P3 has been related to an orienting response to motivational significant stimuli [39, 40]. In addition, Waller et al. [41] found evidence for the P3 to index a generic motor inhibition system, which is recruited in all infrequent events. In the current study, students with a fixed mindset may therefore see the stop stimulus as more motivationally significant, resulting in stronger attentional or inhibitory processing. Note, however, that this finding did not reach the significance threshold and therefore should be interpreted with caution, and needs to be replicated.

Although the uncorrected analysis of the subsequent error-monitoring process demonstrated a significant effect of mindset on Pe, this was rendered non-significant after correcting for the preceding stimulus-evoked overlap. This supports the hypothesis that the Pe is confounded by a delayed P3, especially considering that the direction of the relations between P3 and mindset, and between Pe and mindset, were similar; both P3 and Pe amplitudes were higher for those endorsing more of a fixed mindset (although for the P3 this reached near-significance). Therefore, the uncorrected larger Pe amplitudes in fixed-minded students do not seem to reflect an adaptive mechanism, rather a delayed P3 effect.

Both our uncorrected and corrected findings are difficult to reconcile with two previous findings. First, other studies investigating mindset as trait, found smaller Pe amplitudes in both fixed-minded children and adults [21, 22], which were related to post-error accuracy. Although these studies, both from the same research group, may have valid findings on themselves (but see discussion about potential confounding overlap), we could evidently not replicate them. The inconsistencies may be actually very interesting and suggest that the direction of the effect may depend on (often unknown) contextual factors. One contextual factor that may have influenced our results is that the stop-task was preceded by a more stress-inducing arithmetic task [based on 42] that involved errors as well. This may have negatively primed students with a fixed mindset for errors and performance goals, with carryover effects to the stop-task. According to the motivational intensity theory [43], participants primed to be in a negative mood (sad or fearful) demonstrate more effort during an objectively easy task, due to experiencing higher subjective difficulty. The stop-task can be seen as an objectively easy task, being rather monotone, which may have led negatively primed fixed-minded students to exert more effort (as reflected in enhanced P3/Pe). Future ERP research should aim to find the boundary conditions of when fixed-minded people shift from more effort (related to more emotional/motivational salience) to disengagement (possibly related to avoidance strategies).

Second, posthoc analyses in Schroder et al. [21] did not demonstrate significant correlations between mindset and Pe (incorrect) and Pc (correct) separately, only for the difference wave. In contrast, in our uncorrected analysis, we not only found a significant correlation between mindset and the difference wave, but surprisingly also a positive relationship between mindset and Pc. The larger Pc in growth-minded students, may show that they were more sensitive to positive internal feedback (correct go response). Note that Pc has no preceding stop stimulus, and therefore there is no concern of confounding overlap.

A potential limitation of error-monitoring studies in general is the potential overlap of stimulus- and response-related processing, due to the adjacency between stimulus and response, which may confound error-monitoring ERPs [27, 32]. After re-analyzing the error-monitoring ERPs correcting for this overlap, the relation between Pe and mindset disappeared in the current study. This may be consequential for the mindset ERP literature. It would be less problematic when mindset is not related to stimulus processing, as potential bias would be equal across the mindset continuum; however, in previous studies [23], effects of mindset on preceding stimulus P3 show otherwise, while other studies did not explore ERPs related to the stimuli preceding the errors [21, 22]. Moreover, rare events like stop stimuli and erroneous responses recruit similar brain networks associated with inhibition and re-orienting [44], which makes the overlap even more problematic. At least in the current study, the evidence suggests that mindset differences in error-monitoring were delayed P3 mindset differences related to stop-stimulus processing.

## Conclusions and future directions

We found comparable Ne, but larger Pe amplitudes in fixed-minded students; however, after overlap correction, the Pe results were rendered non-significant. A likely explanation for this overlap was a near-significant effect of mindset on the preceding stimulus P3. Future studies are advised to explore stimulus processing as well, and if needed, to correct error-monitoring ERPs for preceding stimulus overlap. An unexpected finding was a larger Pc (positivity after correct go responses) in growth-minded students, potentially because they were more sensitive to positive internal feedback. Future studies may conduct similar tests for Pc and Pe separately, not only focusing on Pe difference waves (Pe-Pc). Finally, although N2 was larger for trials following errors, mindset was unrelated. Despite this null result, it may still be interesting for

future studies to look into the neural correlates of cognitive control during remedial action (PES/PEA) in association with mindset, as it may play a role depending on the particular context or population.

The current study contributes to the existing literature by hinting at a more complex relationship between mindset and error-monitoring that needs to be reconsidered. Future neuroscience studies can contribute to refining theoretical models of mindset by exploring potential contextual influences and individual variation. This fits in recent attempts by other researchers to investigate more nuanced relationships between mindset and associated constructs, such as achievement goals, using latent profile analysis [45]. It would be particularly interesting to associate different mindset profiles to neurocognitive processes, such as error-monitoring. Another still open question, is whether neurocognitive processes are sensitive to mindset interventions, in addition to brief mindset manipulations. Randomized controlled trials that include such outcomes could add valuable insights into both the causal effects of mindset, and the underlying neural (treatment) mechanisms.

## Author Contributions

**Conceptualization:** Tieme W. P. Janssen, Smiddy Nieuwenhuis, Nienke van Atteveldt.

**Data curation:** Tieme W. P. Janssen.

**Funding acquisition:** Nienke van Atteveldt.

**Investigation:** Tieme W. P. Janssen, Jamie Hoefakker.

**Methodology:** Tieme W. P. Janssen.

**Project administration:** Tieme W. P. Janssen, Jamie Hoefakker, Patricia D. Dreier Gligoor.

**Supervision:** Tieme W. P. Janssen, Patricia D. Dreier Gligoor, Milene Bonte, Nienke van Atteveldt.

**Validation:** Tieme W. P. Janssen.

**Writing – original draft:** Tieme W. P. Janssen, Smiddy Nieuwenhuis, Nienke van Atteveldt.

**Writing – review & editing:** Tieme W. P. Janssen, Jamie Hoefakker, Patricia D. Dreier Gligoor, Milene Bonte, Nienke van Atteveldt.

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
