## [Decision Letter · Decision Letter 0]

6 May 2021

PONE-D-21-00634

Neural correlates of error-monitoring and mindset: back to the drawing board?

PLOS ONE

Dear Dr. Janssen,

Thank you for submitting your manuscript to PLOS ONE. After careful consideration, we feel that it has merit but does not fully meet PLOS ONE’s publication criteria as it currently stands. Therefore, we invite you to submit a revised version of the manuscript that addresses the points raised during the review process.

We look forward to receiving your revised manuscript.

Kind regards,

Kiyoshi Nakahara, PhD

Academic Editor

PLOS ONE

Journal Requirements:

3) We note that you have stated that you will provide repository information for your data at acceptance. Should your manuscript be accepted for publication, we will hold it until you provide the relevant accession numbers or DOIs necessary to access your data. If you wish to make changes to your Data Availability statement, please describe these changes in your cover letter and we will update your Data Availability statement to reflect the information you provide.

Reviewers' comments:

Reviewer's Responses to Questions

**Comments to the Author**

1. Is the manuscript technically sound, and do the data support the conclusions?

Reviewer #1: Yes

Reviewer #2: Yes

2. Has the statistical analysis been performed appropriately and rigorously? 

Reviewer #1: I Don't Know

Reviewer #2: Yes

3. Have the authors made all data underlying the findings in their manuscript fully available?

Reviewer #1: Yes

Reviewer #2: Yes

4. Is the manuscript presented in an intelligible fashion and written in standard English?

Reviewer #1: Yes

Reviewer #2: Yes

5. Review Comments to the Author

Reviewer #1: The paper compares ERPs across participants and evaluates the effect of mindset on ERPs. An interference effect of the P3 on the errP is reported.

The study is interesting and has some considerable relevance to the study of error related potentials as a function on mindsets. I have some comments and requests for the authors.

Introduction: I would like to see a broader discussion of mindsets and effects on behaviour and the brain, including during mathematics tasks. Some papers that may prove useful are listed below…

O’Rourke E, Haimovitz K, Ballweber C, Dweck CS, Popovíc Z. Brain points: A growth mindset incentive structure boosts persistence in an educational game. Conf. Hum. Factors Comput. Syst. - Proc., New York, NY, USA: Association for Computing Machinery; 2014, p. 3339–48. https://doi.org/10.1145/2556288.2557157.

Sarrasin JB, Nenciovici L, Foisy LMB, Allaire-Duquette G, Riopel M, Masson S. Effects of teaching the concept of neuroplasticity to induce a growth mindset on motivation, achievement, and brain activity: A meta-analysis. Trends Neurosci Educ 2018;12:22–31. https://doi.org/10.1016/j.tine.2018.07.003.

Daly I, Bourgaize J, Vernitski A. Mathematical mindsets increase student motivation: Evidence from the EEG. Trends Neurosci Educ 2019;15:18–28. https://doi.org/10.1016/J.TINE.2019.02.005.

Ng B. The Neuroscience of Growth Mindset and Intrinsic Motivation. Brain Sci 2018;8:20.

https://doi.org/10.3390/brainsci8020020.

Analysis: Please add more details on the participant’s task.

Analysis: Please provide full details of all analysis steps, as many readers will not want to locate and read additional papers to understand the analysis process (e.g. see line 194).

Analysis: How were ‘artifact free segments’ identified (line 203)?

Analysis: The criteria for choosing the ERP time segments (lines 224-237) is not clear. This seems to be based on some statistical measure, however the exact test used is not specified?

I am concerned that the time window search followed by the significance test is somewhat akin to “double dipping”. The current approach appears to be that (1) the entire dataset from all 89 participants is used to identify optimal time windows, then (2) the effect of mindset on these time windows is measured, again on the same 89 participants. If this is the case then the approach is essentially akin to double dipping (or similar to a form of p-hacking). I suggest that instead the authors adopt a machine learning approach and first use some portion of the dataset (perhaps half) to identify the optimal time windows, then test the effect of participant mindsets on these time windows using the other half of the dataset. This is equivalent to the approach suggested here …

Kriegeskorte N, Simmons WK, Bellgowan PS, Baker CI. Circular analysis in systems neuroscience: The dangers of double dipping. Nat Neurosci 2009;12:535–40. https://doi.org/10.1038/nn.2303.

… of course if I’ve misunderstood the approach then perhaps a little more clarity is needed in the text to explain the analysis approach.

Reviewer #2: A suggestion is to strengthen the research contribution of the paper and to have a conclusion section after Discussion section (although the authors put “conclusion” as last paragraph of Discussion section).

6. PLOS authors have the option to publish the peer review history of their article (what does this mean?). If published, this will include your full peer review and any attached files.

Reviewer #1: No

Reviewer #2: No

---

## [Author Response · Author response to Decision Letter 0]

4 Jun 2021

Dear editor and reviewers,

We would like to thank you and the reviewers for the helpful comments, and thank you for giving us the opportunity to revise and resubmit our manuscript. We think that the comments have been very helpful to improve the readability and quality of our manuscript. In the rebuttal note, we provide details on how we have incorporated the feedback into our manuscript. We feel that all issues have been resolved satisfactory in the current version of the manuscript and look forward to your decision on publication.

On behalf of the authors.

Kind regards,

Dr. Tieme Janssen

Editor:

Response: We have checked the reference list.

Response: We checked the requirements.

3) We note that you have stated that you will provide repository information for your data at acceptance. Should your manuscript be accepted for publication, we will hold it until you provide the relevant accession numbers or DOIs necessary to access your data. If you wish to make changes to your Data Availability statement, please describe these changes in your cover letter and we will update your Data Availability statement to reflect the information you provide.

Response: We will provide repository information in the case this manuscript is accepted.

Reviewer #1: 

The paper compares ERPs across participants and evaluates the effect of mindset on ERPs. An interference effect of the P3 on the errP is reported.

The study is interesting and has some considerable relevance to the study of error related potentials as a function on mindsets. I have some comments and requests for the authors.

1. Introduction: I would like to see a broader discussion of mindsets and effects on behaviour and the brain, including during mathematics tasks. Some papers that may prove useful are listed below…

O’Rourke E, Haimovitz K, Ballweber C, Dweck CS, Popovíc Z. Brain points: A growth mindset incentive structure boosts persistence in an educational game. Conf. Hum. Factors Comput. Syst. - Proc., New York, NY, USA: Association for Computing Machinery; 2014, p. 3339–48. https://doi.org/10.1145/2556288.2557157.

Sarrasin JB, Nenciovici L, Foisy LMB, Allaire-Duquette G, Riopel M, Masson S. Effects of teaching the concept of neuroplasticity to induce a growth mindset on motivation, achievement, and brain activity: A meta-analysis. Trends Neurosci Educ 2018;12:22–31. https://doi.org/10.1016/j.tine.2018.07.003.

Daly I, Bourgaize J, Vernitski A. Mathematical mindsets increase student motivation: Evidence from the EEG. Trends Neurosci Educ 2019;15:18–28. https://doi.org/10.1016/J.TINE.2019.02.005.

Ng B. The Neuroscience of Growth Mindset and Intrinsic Motivation. Brain Sci 2018;8:20.

https://doi.org/10.3390/brainsci8020020.

Response: We thank the reviewer for these suggestions. Because we want to keep the manuscript focused, we were selective in incorporating some of the suggestions (e.g. Ng. B., 2018) and added a few of ourselves to broaden the discussion. Some of the studies, including the one about mathematical mindset (Daly et al., 2019), we deemed outside the scope of this manuscript.

Page 3, lines 44-51 (italic is added text in the uploaded Response to reviewers document):

“So far, only a few neurocognitive studies have investigated the neural mechanism underlying mindset [2], typically by investigating error and feedback processing. These studies yielded some common findings, but also inconsistencies, open questions and methodological limitations. Neuroscience research has the potential to refine theoretical models of non-cognitive skills, such as mindset, and may contribute to developing well-informed interventions [3]. This is especially important considering recent discussions about the foundations of mindset theory [4] and the relevance of mindset interventions [5,6].” 

In addition, we broadened the discussion about the brain correlates, now also including the neural generators involved in the Ne and Pe components, and by including relevant fMRI studies into mindset.

Page 3, lines 53-56:

“Ne is a response-locked ERP that reaches a negative fronto-central maximum around 50-100ms after the initiation of an erroneous response [7], and is presumably generated in the anterior cingulate cortex [ACC; 8].”

Page 3, lines 58-61:

“Unlike the Ne, the neural generators of the Pe are less well known, and are presumably more distributed [10], including the ACC [11], posterior cingulate cortex [12,13], and the parietal and insular cortices [14,15].”

Page 5, lines 86-91:

“This conclusion is corroborated by a resting-state fMRI study [3], which demonstrated growth mindset to be related to increased connectivity between brain regions that support regulation strategies and error monitoring, comprising the dorsal striatum, dorsal ACC and left dorso-lateral prefrontal cortex (DLPFC). Another fMRI study relevant to mindset, showed mastery-oriented participants to have relatively increased DLPFC activity during negative feedback compared to performance-approach-oriented participants [25].”

Added References

2. Ng B. The neuroscience of growth mindset and intrinsic motivation. Brain Sci. 2018;8. doi:10.3390/brainsci8020020

3. Myers CA, Wang C, Black JM, Bugescu N, Hoeft F. The matter of motivation: Striatal resting-state connectivity is dissociable between grit and growth mindset. Soc Cogn Affect Neurosci. 2016;11: 1521–1527. doi:10.1093/scan/nsw065

4. Burgoyne AP, Hambrick DZ, Macnamara BN. How Firm Are the Foundations of Mind-Set Theory? The Claims Appear Stronger Than the Evidence. Psychol Sci. 2020;31: 258–267. doi:10.1177/0956797619897588

5. Sisk VF, Burgoyne AP, Sun J, Butler JL, Macnamara BN. To What Extent and Under Which Circumstances Are Growth Mind-Sets Important to Academic Achievement? Two Meta-Analyses. Psychol Sci. 2018;29: 549–571. doi:10.1177/0956797617739704

6. Miller DI. When Do Growth Mindset Interventions Work? Trends Cogn Sci. 2019;23: 910–912. doi:10.1016/j.tics.2019.08.005

8. Bush G, Luu P, Posner MI. Cognitive and emotional influences in anterior cingulate cortex. Trends Cogn Sci. 2000;4: 215–222. doi:10.1016/S1364-6613(00)01483-2

10. Wessel JR. Error awareness and the error-related negativity: evaluating the first decade of evidence. Front Hum Neurosci. 2012;6: 1–16. doi:10.3389/fnhum.2012.00088

11. Herrmann MJ, Römmler J, Ehlis AC, Heidrich A, Fallgatter AJ. Source localization (LORETA) of the error-related-negativity (ERN/Ne) and positivity (Pe). Cogn Brain Res. 2004;20: 294–299. doi:10.1016/j.cogbrainres.2004.02.013

12. Vocat R, Pourtois G, Vuilleumier P. Unavoidable errors: A spatio-temporal analysis of time-course and neural sources of evoked potentials associated with error processing in a speeded task. Neuropsychologia. 2008;46: 2545–2555. doi:10.1016/j.neuropsychologia.2008.04.006

13. O’Connell RG, Dockree PM, Bellgrove MA, Kelly SP, Hester R, Garavan H, et al. The role of cingulate cortex in the detection of errors with and without awareness: A high-density electrical mapping study. Eur J Neurosci. 2007;25: 2571–2579. doi:10.1111/j.1460-9568.2007.05477.x

14. Orr C, Hester R. Error-related anterior cingulate cortex activity and the prediction of conscious error awareness. Front Hum Neurosci. 2012;6: 1–12. doi:10.3389/fnhum.2012.00177

15. Veen V van, Carter CS. The Timing of Action-Monitoring Processes in the Anterior Cingulate Cortex. J Cogn Neurosci. 2002;14: 593–602. doi:10.1162/08989290260045837

25. Lee W, Kim S Il. Effects of achievement goals on challenge seeking and feedback processing: Behavioral and fMRI evidence. PLoS One. 2014;9. doi:10.1371/journal.pone.0107254

2. Analysis: Please add more details on the participant’s task.

Response: The participant’s task, which was a stop-signal-task (SST), is explained on page 7, lines 157-180. In addition, Figure 1 shows a graphical representation of the task stimuli.

3. Analysis: Please provide full details of all analysis steps, as many readers will not want to locate and read additional papers to understand the analysis process (e.g. see line 194).

Response: Thank you for the suggestion. We added more details:

Page 10, lines 209-210:

“Trials were segmented from -100ms to 800ms relative to the stop stimulus. Subsequently, a 100ms pre-stimulus baseline was applied and averages were obtained for SS.”

4. Analysis: How were ‘artifact free segments’ identified (line 203)?

Response: Please see page 9, lines 194-198:

“Ocular artifacts were estimated and corrected with semi-automatic independent component analysis [31], and automatic artifact rejection was applied to segments based on the following criteria: maximum allowed voltage step of 50µV/ms, maximal peak-to-peak amplitude difference of ±150µV, and minimal low activity of 0.50µV for 100ms intervals.”

5. Analysis: The criteria for choosing the ERP time segments (lines 224-237) is not clear. This seems to be based on some statistical measure, however the exact test used is not specified?

Response: Thank you for noting that the procedure was unclear. First, we did not use any statistical measure to define the windows. Rather, we used an automatic search in BrainVision Analyzer for the largest amplitude in the grand average ERP waveforms (or difference waves) to define the peak, and then defined a symmetrical window of either 50 ms or 100 ms (smaller window for earlier and smaller components, larger window for later larger/extended components – see Luck 2014, Chapter 9) around the peak to obtain mean amplitudes. Importantly, these windows were compared to the existing literature, to ascertain their correct identification.

We added more information, also in response to point 6 raised by the reviewer (see pages 11-12, lines 240-258; italic is added text):

“Based on all 89 participants, irrespective of their reported mindset, grand average ERPs, scalp topographies and difference waves for each trail type were inspected to define analysis windows (see Figs 2, 3 and 6). In addition, characteristics of the ERP waveforms were compared to the literature [18,34–36], to ascertain the correct identification of ERP components. This resulted in the following time windows: inhibition P3 (336-386ms), error monitoring Ne (77-127ms) and Pe (203-403ms), cognitive control N2 (217-276ms) and, unplanned, P3 (310-410ms). The median split in Figs 2 and 3, with a growth and fixed mindset group, is only for visualization purposes and did not inform the selected time windows.

 More specifically, all windows were symmetrically centered based on the latency of the electrode with the largest amplitude. Ne and Pe windows were based on difference waves (incorrect-correct response). Mean voltage amplitudes were extracted and used for statistical analyses. To validate the condition effects for error-monitoring, besides exporting Ne and Pe difference waves, also the Nc/Pc [correct] and Ne/Pe [incorrect] were exported separately. A minimum of 20 artefact-free segments were required for each condition to be included. Due to the task design, there were more correct than incorrect responses (SE) contributing to their respective ERPs, resulting in different signal-to-noise ratio’s. However, when using mean rather than peak amplitudes, it will not bias the results and it is recommended to retain all trials to decrease Type II error rate [35].”

6. I am concerned that the time window search followed by the significance test is somewhat akin to “double dipping”. The current approach appears to be that (1) the entire dataset from all 89 participants is used to identify optimal time windows, then (2) the effect of mindset on these time windows is measured, again on the same 89 participants. If this is the case then the approach is essentially akin to double dipping (or similar to a form of p-hacking). I suggest that instead the authors adopt a machine learning approach and first use some portion of the dataset (perhaps half) to identify the optimal time windows, then test the effect of participant mindsets on these time windows using the other half of the dataset. This is equivalent to the approach suggested here …

Kriegeskorte N, Simmons WK, Bellgowan PS, Baker CI. Circular analysis in systems neuroscience: The dangers of double dipping. Nat Neurosci 2009;12:535–40. https://doi.org/10.1038/nn.2303.

… of course if I’ve misunderstood the approach then perhaps a little more clarity is needed in the text to explain the analysis approach.

Response: We respectfully disagree with the suggestion that there might be double dipping or p-hacking, but we do agree that our approach can be explained more clearly. Double dipping would have been the case when the time windows were based on maximal (visual) differences between the fixed and growth mindset groups. That is however not the case here. Based on the grand average of all 89 participants we selected time windows for the various ERP components, this is irrespective of our variable of interest: mindset. Therefore, we could not have maximized our chances of finding a relationship with mindset. We do understand the confusion, as Figures 2 and 3 show separate ERPs for a growth mindset and fixed mindset group, based on a median split. However, this was only for visualisation purposes, as stated on page 12 (lines 265-266) of the manuscript, not for statistical purposes. In addition, the role of mindset was only statistically tested at the very end, when adding the corresponding variable as covariate to the analysis.

We changed several descriptions in the revised manuscript to better explain our approach, and to avoid the suggestion of double dipping. First we emphasize that mindset did not play a role in window selection, and in addition, we now added that validating/comparing our ERP waveforms to the literature was part of the approach as well (previously this was only reported in the results section – see Luck, 2014, chapter 9, on quantification of ERP waveforms):

Page 10, see pages 11-12, lines 240-258:

“Based on all 89 participants, irrespective of their reported mindset, grand average ERPs, scalp topographies and difference waves for each trail type were inspected to define analysis windows (see Figs 2, 3 and 6). In addition, characteristics of the ERP waveforms were compared to the literature [18,34–36], to ascertain the correct identification of ERP components. This resulted in the following time windows: inhibition P3 (336-386ms), error monitoring Ne (77-127ms) and Pe (203-403ms), cognitive control N2 (217-276ms) and, unplanned, P3 (310-410ms). The median split in Figs 2 and 3, with a growth and fixed mindset group, is only for visualization purposes and did not inform the selected time windows.

 More specifically, all windows were symmetrically centered based on the latency of the electrode with the largest amplitude. Ne and Pe windows were based on difference waves (incorrect-correct response). Mean voltage amplitudes were extracted and used for statistical analyses. To validate the condition effects for error-monitoring, besides exporting Ne and Pe difference waves, also the Nc/Pc [correct] and Ne/Pe [incorrect] were exported separately. A minimum of 20 artefact-free segments were required for each condition to be included. Due to the task design, there were more correct than incorrect responses (SE) contributing to their respective ERPs, resulting in different signal-to-noise ratio’s. However, when using mean rather than peak amplitudes, it will not bias the results and it is recommended to retain all trials to decrease Type II error rate [35].”

Reviewer #2: 

1. A suggestion is to strengthen the research contribution of the paper and to have a conclusion section after Discussion section (although the authors put “conclusion” as last paragraph of Discussion section).

Response: We thank the reviewer for the suggestions. First, we extended the discussion in the paragraph where we compare our results to previous studies:

Page 20, lines 447-461:

“Although these studies, both from the same research group, may have valid findings on themselves (but see discussion about potential confounding overlap), we could evidently not replicate them. The inconsistencies may be actually very interesting and suggest that the direction of the effect may depend on (often unknown) contextual factors. One contextual factor that may have influenced our results is that the stop-task was preceded by a more stress-inducing arithmetic task [based on 42] that involved errors as well. This may have negatively primed students with a fixed mindset for errors and performance goals, with carryover effects to the stop-task. According to the motivational intensity theory [43], participants primed to be in a negative mood (sad or fearful) demonstrate more effort during an objectively easy task, due to experiencing higher subjective difficulty. The stop-task can be seen as an objectively easy task, being rather monotone, which may have led negatively primed fixed-minded students to exert more effort (as reflected in enhanced P3/Pe). Future ERP research should aim to find the boundary conditions of when fixed-minded people shift from more effort (related to more emotional/motivational salience) to disengagement (possibly related to avoidance strategies).”

In addition we added a ‘Conclusions and future directions’ section at the end of the discussion, where we emphasized our research contribution to the literature, and proposed future research based on our findings:

Page 22, lines 484-507:

“Conclusions and future directions

 We found comparable Ne, but larger Pe amplitudes in fixed-minded students; however, after overlap correction, the Pe results were rendered non-significant. A likely explanation for this overlap was a near-significant effect of mindset on the preceding stimulus P3. Future studies are advised to explore stimulus processing as well, and if needed, to correct error-monitoring ERPs for preceding stimulus overlap. An unexpected finding was a larger Pc (positivity after correct go responses) in growth-minded students, potentially because they were more sensitive to positive internal feedback. Future studies may conduct similar tests for Pc and Pe separately, not only focusing on Pe difference waves (Pe-Pc). Finally, although N2 was larger for trials following errors, mindset was unrelated. Despite this null result, it may still be interesting for future studies to look into the neural correlates of cognitive control during remedial action (PES/PEA) in association with mindset, as it may play a role depending on the particular context or population.

 The current study contributes to the existing literature by hinting at a more complex relationship between mindset and error-monitoring that needs to be reconsidered. Future neuroscience studies can contribute to refining theoretical models of mindset by exploring potential contextual influences and individual variation. This fits in recent attempts by other researchers to investigate more nuanced relationships between mindset and associated constructs, such as achievement goals, using latent profile analysis [45]. It would be particularly interesting to associate different mindset profiles to neurocognitive processes, such as error-monitoring. Another still open question, is whether neurocognitive processes are sensitive to mindset interventions, in addition to brief mindset manipulations. Randomized controlled trials that include such outcomes could add valuable insights into both the causal effects of mindset, and the underlying neural (treatment) mechanisms.”

---

## [Decision Letter · Decision Letter 1]

24 Jun 2021

Neural correlates of error-monitoring and mindset: back to the drawing board?

PONE-D-21-00634R1

Dear Dr. Janssen,

We’re pleased to inform you that your manuscript has been judged scientifically suitable for publication and will be formally accepted for publication once it meets all outstanding technical requirements.

Kind regards,

Kiyoshi Nakahara, PhD

Academic Editor

PLOS ONE

Additional Editor Comments (optional):

Reviewers' comments:

Reviewer's Responses to Questions

**Comments to the Author**

1. If the authors have adequately addressed your comments raised in a previous round of review and you feel that this manuscript is now acceptable for publication, you may indicate that here to bypass the “Comments to the Author” section, enter your conflict of interest statement in the “Confidential to Editor” section, and submit your "Accept" recommendation.

Reviewer #1: All comments have been addressed

Reviewer #2: All comments have been addressed

2. Is the manuscript technically sound, and do the data support the conclusions?

Reviewer #1: Yes

Reviewer #2: (No Response)

3. Has the statistical analysis been performed appropriately and rigorously? 

Reviewer #1: Yes

Reviewer #2: (No Response)

4. Have the authors made all data underlying the findings in their manuscript fully available?

Reviewer #1: Yes

Reviewer #2: (No Response)

5. Is the manuscript presented in an intelligible fashion and written in standard English?

Reviewer #1: Yes

Reviewer #2: (No Response)

6. Review Comments to the Author

Reviewer #1: The authors have addressed all my concerns effectively.

(some additional characters to meet the 100 char min)

Reviewer #2: (No Response)

7. PLOS authors have the option to publish the peer review history of their article (what does this mean?). If published, this will include your full peer review and any attached files.

Reviewer #1: No

Reviewer #2: No

---

## [Editor Report · Acceptance letter]

20 Jul 2021

PONE-D-21-00634R1 

Neural correlates of error-monitoring and mindset: back to the drawing board? 

Dear Dr. Janssen:

I'm pleased to inform you that your manuscript has been deemed suitable for publication in PLOS ONE. Congratulations! Your manuscript is now with our production department. 

Kind regards, 

on behalf of

Dr. Kiyoshi Nakahara 

Academic Editor

PLOS ONE